

# Transfer Entropy between South Atlantic Anomaly and Global Sea Level for the last 300 years

Saioa A. Campuzano[1,2], Angelo De Santis[3,4], Francisco Javier Pavón-Carrasco[1], María Luisa Osete[1,2], Enkelejda Qamili[5]

[1]Dpto. de Física de la Tierra, Astronomía y Astrofísica I, Universidad Complutense de Madrid (UCM), Avd. Complutense s/n, 28040-Madrid, Spain
[2]Instituto de Geociencias (IGEO) CSIC, UCM, Ciudad Universitaria, 28040-Madrid, Spain
[3]Istituto Nazionale di Geofisica e Vulcanologia (INGV), Via Vigna Murata, 605, 00143-Roma, Italy
[4]Dipartimento di Ingegneria e Geologia, Università degli Studi "G. D'Annunzio", Chieti, Italy
[5]Serco spa. Frascati (Rome), Italy

*Correspondence to*: S. A. Campuzano (sacampuzano@ucm.es)

**Abstract.** An innovative information-theoretic tool, transfer entropy, has been applied to measure the possible information flow and sense between two real time series: the South Atlantic Anomaly (SAA) area extent at the Earth's surface and the Global Sea Level (GSL) rise anomalies for the last 300 years. This connection was previously suggested considering only the long term trend. Now we study the possibility of that this relation also happens in shorter scales. The new results seem to support again this hypothesis, with more information transferred from the SAA to the GSL anomalies, with about 90% of confidence level. This could provide a new clue on the existence of a link between the geomagnetic field and the Earth's climate in the past.

## 1 Introduction

The possible relationship between the Earth's climate and geomagnetic field has been highly debated in the last fifty years [e.g. Wollin et al., 1971; Wagner et al., 2001; Christl et al., 2004; Gallet et al., 2005; Courtillot et al., 2007; Thouveny et al., 2008; Knudsen and Riisager, 2009; Kitaba et al., 2013; Rossi et al., 2014] but it is still an open question. The first serious proposal that quantifies this possible link was given by Wollin et al. [1971], who pointed out that low geomagnetic intensities are generally associated with warm climate periods (similar to the current situation), and by Bucha [1976] who suggested that drifts of geomagnetic poles could have been responsible for displacements of a large low-pressure region of the Earth's atmosphere associated with an increase of cyclonic activity and sudden climate changes [Bucha, 1978].

Throughout the last few decades, other mechanisms that could explain the geomagnetic field-climate relation have been proposed. The most plausible at long-time scale is related to the rate of galactic cosmic rays coming to the Earth's surface. This flux of galactic cosmic rays is modulated by the intensity of both Sun and the Earth's magnetic fields that act as a protective shield. High values of the solar (and Earth's) magnetic field intensity reinforce the shield and then a low density of galactic cosmic rays coming to the Solar System (and in turn to Earth) is expected [see, for example, Snowball and Mucheler,



2007]. Entering the atmosphere, the cosmic rays could play an important role in cloud formation [Duplissy et al., 2010; Kirkby et al., 2011] and, in this way the geomagnetic field would be involved in climate processes. That is, a decreasing in the geomagnetic field intensity would allow a higher entrance of galactic cosmic rays to the Earth which could enhance the formation of low-lying clouds [Svensmark and Friis-Christensen, 1997; Svensmark, 1998; Usoskin and Kovaltsov, 2008] or

increase the global cloud cover leading to tropospheric cooling [Christl et al., 2004]. This mechanism was invoked to explain the possible relation between the intensity of Earth's magnetic field and climate on glacial-interglacial timescales, since dipole moment lows (related to geomagnetic excursions) seem to occur shortly before the onset of relatively cold intervals [Thouveny et al., 2008; Kitaba et al., 2013]. This suggests a connection between low geomagnetic intensity and climatic cooling. However, such connection could be circumstantial, as pointed out by these authors, since the variations in geomagnetic field intensity

may, in fact, be linked to variations in Earth's orbital parameters [Thouveny et al., 2008], which are considered the main climate-controlling factors in the Pleistocene [Hays et al., 1976].

On the other hand, Gallet et al. [2005] compared the advance and retreat of the Alpine Glaciers during the last three millennia with increases and decreases of the geomagnetic field intensity in Paris estimated from archaeomagnetic data (palaeomagnetic data from heated archaeological artefacts). A later work with a more complete palaeomagnetic intensity

database corroborated a similar connection at European continental scale [Pavón-Carrasco et al., 2008]. The results of these studies suggest a possible link between centennial-scale cooling episodes and enhanced geomagnetic intensity, the opposite to the galactic cosmic rays mechanism [Svensmark, 1998; Christl et al., 2004; Usoskin and Kovaltsov, 2008; Thouveny et al., 2008; Kitaba et al., 2013] but in agreement with the first links established in the 70's by Bucha [1976, 1978] and Wollin et al. [1971].

Other studies point out other possible mechanisms which explain this connection, such as the experimental result of Pazur and Winklhofer [2008]. They focus on the effect of the geomagnetic intensity on $CO_2$ solubility in the ocean. They observed that low values of geomagnetic field intensity reduce the $CO_2$ solubility in the ocean, displacing more $CO_2$ to the atmosphere and increasing the temperature.

All this leads to the deduction that the possible link between the Earth's climate and the geomagnetic field is far from

being demonstrated and understood.

For shorter time scales, i.e. last 300 years, De Santis et al. [2012; 2013] observed a similar temporal trend between the growing South Atlantic Anomaly (SAA) area extent on the Earth's surface and the Global Sea Level (GSL) rise. The SAA is one of the most outstanding features of the geomagnetic field. It is a large geomagnetic anomaly, presently covering a large area over the Western coast of Africa, the South Atlantic Ocean, the major part of South America and the South-western Pacific

Ocean, which reaches lower values of intensity than expected at those geomagnetic latitudes. Several studies [Gubbins, 1987; Hulot et al., 2002; Olson and Amit, 2006; Gubbins et al., 2006; Finlay, 2012] point out that this anomaly is the response on the Earth's surface of a reverse flux path located at the terrestrial CMB (core-mantle boundary).

In this work, we propose to study, for the first time, the possible causal information link between the anomalies of SAA surface extent and GSL rise for the last 300 years by means of an innovative statistical tool for non-linear dynamic studies



which measures the information flux and the sense of this flux between two systems (for example, two real time series): the Transfer Entropy (TE; Schreiber, 2000). We will show a predominant information flux from SAA to GSL anomalies, and present some possible physics mechanisms recently proposed to explain it.

The present paper is structured as follows: in the first section we expose the chosen time series to carry out this
analysis. Then, we explain the details on the main methodologies applied in this work. Finally, in the discussion and conclusions we summarize the outcomes reached and their possible future implications.

## 2 Data

We analyze two time series: a) the SAA area extent at the Earth's surface given by different geomagnetic models (GUFM1 model, Jackson et al., 2000; and the later modifications given by Gubbins et al., 2006 and Finlay, 2008), and b) the
GSL reconstruction for the last 300 years [Jevrejeva et al., 2008]. Both time series are detailed below.

The SAA surface extent has been computed from the three mentioned historical geomagnetic models covering the last 400 years. The difference between these models lies in the method used to estimate the first Gauss coefficient ($g_1^0$) prior to 1840 AD, due to the lack of instrumental intensity data before that year. Jackson et al. [2000] extrapolated linearly the value of this coefficient backwards from 1840 and they assumed a constant rate of temporal evolution of 15 nT/yr, which corresponds
to the average time rate of $g_1^0$ from 1850 to 1990. Gubbins et al. [2006] modified the $g_1^0$ by using the Korte et al. [2005] intensity palaeomagnetic database for the period from 1590 to 1840 to obtain a more realistic value of this coefficient. And more recently, Finlay [2008], using the same palaeomagnetic database of Gubbins et al. [2006], applied different statistic approaches to fix again the coefficient $g_1^0$ providing no rate of change for that coefficient from 1590 to 1840. Consequently the estimations of the SAA surface extent obtained by these models differ slightly for times prior to 1840, but agree for the
most recent period (see Fig. 1a). The SAA surface extent could be defined, in practice, by the area below a given intensity contour line at the Earth's surface (here we selected the contour line of 32000 nT following De Santis et al., 2012).

For the Global mean Sea Level (GSL), we use a reconstruction since 1700 based on the longest available tide-gauge records [Jevrejeva et al., 2008; http://www.psmsl.org/products/reconstructions/jevrejevaetal2008.php], where the effects of vertical land movement induced by the glacial isostatic adjustment of the solid Earth have been removed. Jevrejeva et al.
[2008] extended the record backwards from 1850 using three of the longest (though discontinuous) tide gauge records available, being the error of the reconstruction higher in this epoch (Fig. 1b).

We have smoothed both the SAA and the GSL series by using penalized cubic splines in order to avoid future mathematical artefacts resulting from the differences in the reconstruction prior and after 1850. For both records, the fitting was carried out using knot points every 5 years from 1700 to 2000 and a spline damping parameter of 10 $yr^4/km^4$ and 10
$yr^4/mm^2$ for the SAA and GSL time series, respectively. These optimal values were estimated according to the root mean square (rms) error (see Fig. S1 in the Supplementary Material).

In general, the Transfer Entropy (TE) is applied on stationary time series [e.g. Marschinski and Kantz, 2002]. However, as evident from Fig. 1, both SAA and GSL series cannot reasonably be assumed as stationary, being both curves

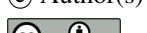



almost monotonically increasing. For this reason, we will apply the TE to the anomaly time series after removing the best-fit long-period trend (see Fig. 2). In our case, we choose the simplest polynomial function which accounts for the time evolution of the series: a second order polynomial, which seems the best compromise to remove a reasonable trend and not to completely destroy some similar short-period fluctuations in both series. So, a positive/negative anomaly means that the SAA area extent

or GSL rise grow more/lesser than expected.

**3 Methodology**

TE is an information theoretic measure introduced by Schreiber [2000] as a generalization of the mutual information [Shannon, 1948]. While the mutual information neither contains dynamics nor directional information, the TE takes into account the dynamics of information transport between two systems. This allows quantifying both the exchange of information

to the predominant sense of this flow.

The foundations of the TE are to be found in the basic works of the theory of information [Shannon and Weaver, 1949]. The Shannon entropy is given by:

$$H_I = -\sum_i p(i) log_2 p(i),$$
[1]

where $i$ represents the states that the process $I$ can assume and $p(i)$ the probability distribution which they follow. This quantity measures the average amount of information needed to encode a process optimally.

From finite-order Markov processes, Schreiber [2000] introduced a measure to quantify information transfer between two different time series, based on appropriately conditioned transition probabilities instead of static probabilities. Assuming that the system under study can be approximated by a stationary Markov process of order $k$, the transition probabilities describing the evolution of the system are $p(i_{n+1}|i_n, \ldots, i_{n-k+1})$. If two processes $I$ and $J$ are independent, then the generalized Markov property

$$p(i_{n+1}|i_n, \ldots, i_{n-k+1}) = p\left(i_{n+1}|i_n^{(k)}, j_n^{(l)}\right),$$
[2]

holds, where $i_n^{(k)} = (i_n, \ldots, i_{n-k+1})$, $j_n^{(l)} = (j_n, \ldots, j_{n-l+1})$ and $l$ indicates the number of conditioning states for process $J$.

Schreiber proposed using the Kullback entropy for conditional probabilities [Kullback and Leibler, 1951; Kullback, 1959] to measure the incorrectness of assuming the generalized Markov property (Eq. [2]), i.e. $I$ and $J$ are independent, which results in:

$$TE_{J \to I} = \sum p(i_{n+1}, i_n^{(k)}, j_n^{(l)}) log \frac{p(i_{n+1}, i_n^{(k)}, j_n^{(l)}) p(i_n^{(k)})}{p(i_n^{(k)}, j_n^{(l)}) p(i_{n+1}, i_n^{(k)})},$$
[3]

denoted as transfer entropy (a schematic representation of the TE can be found in Fig. S2). The TE can be understood as the

excess amount of information that must be used to encode the state of a process by erroneously assuming that the actual transition probability distribution function is $p(i_{n+1}|i_n^{(k)})$, instead of $p(i_{n+1}|i_n^{(k)}, j_n^{(l)})$.





There are different strategies to calculate the TE from the analysis of real data. Here, we use the method based on the discretization of the time series, which was explained in details by Sandoval Jr. [2014]. This method consists in dividing the data in a number of bins $S$, by assigning a numeric symbol to each bin from 1 to $S$. Each symbol corresponds to a range of values of data series, which are substituted by the symbols assigned (from 1 to $S$).

Obviously, the calculation of TE will depend on the specific partition chosen $S$. In order to obtain the optimal number of bins $S$, we consider the approach proposed by Knuth [2013], where $S$ is given by the maximization of the posterior probability $p(S|N, n_k)$. Given a uniform bin-width histogram for a statistical data set of $N$ samples, the posterior probability $p(S | N, n_k)$ is given by:

$$p(S|N,n_k) \propto \left(\frac{S}{V}\right)^N \frac{\Gamma(S/2)}{\Gamma(1/2)^S} \frac{\prod_k \Gamma(n_k + 1/2)}{\Gamma(N + S/2)},$$
[4]

where $n_k$ is the number of samples in the $k^{th}$ bin, $V$ is the data range length, and $\Gamma$ is the Gamma function. In optimization
problems, it is common to maximize the logarithm of the Eq. [4] [Knuth, 2013], also because from the behaviour of the logarithm one can study if the chosen time series are sufficiently long to be analyzed with a tool like the TE [Knuth et al., 2005]. For this reason, we maximize the logarithm of the posterior probability to estimate the optimal number of bins.

Once we have checked that the number of data is enough and estimated the optimal number of bins S, we discretize the time series as we explained above, and compute directly the TE from the Eq. [3] given by Schreiber [2000], with $i_n^{(k)}$ and
$j_n^{(l)}$ representing both involved series. The choice of the embedding dimension $k$ and $l$ is a key point in the computation of the TE. If the dimension is too low, the information contained in the past time (or memory) of the series $I$ might be assigned to come from $J$. In order to avoid this, we must get that the series $I$ is independent from itself with a delay $k$. So, we base the selection of this parameter on the determination of the mutual information between the time series $I$ and itself with a delay $k$ [Dimpfl and Peter, 2013]:

$$M_{II_k}(k) = \sum_{i,i_k} p(i,i_k) log \frac{p(i,i_k)}{p(i)p(i_k)},$$
[5]

being $I_k$ the time series $I$ with delay $k$. The value of $k$ associated with the first local minimum reported in the Eq. [5] is considered the optimal embedding dimension.

For the dimension of the embedding $l$ of the $J$ series, it is usually considered $l=1$ or $l=k$ [Schreiber, 2000; Marschinski and Kantz, 2002]. In a conservative approach we consider $l=1$. To calculate the different probabilities of the Eq. [3] we simply
count the number of times that a symbol appears in our time series.

To establish the statistical significance of our results we calculate the TE with the data points of the $J$ series, which represents the source of the presumed information flow, shuffled randomly [see for example Marschinski and Kantz, 2002; Sensoy et al., 2014]. The objective of this procedure is to destroy all potential relations between the two series, $I$ and $J$, and



hence the observed TE should be zero. In finite time series this value rarely is zero due to the finite sample effects, and we obtain a threshold value of TE above which is significant. Practically, we create 1000 surrogate time series of $J$ by using the Iterated Amplitude Adjusted Fourier Transform technique (IAAFT, Theiler et al., 1992; Kugiumtzis, 2000; Schreiber and Schmitz, 2000). This procedure assures that the surrogate time series have the same mean, variance, autocorrelation function

and therefore, power spectrum as the original series. However, we achieve to destroy the non-linear relations and, therefore, the information actually significant transferred from $J$ to $I$ series. To consider the original TE significant, the new TEs, calculated from surrogate series $J$, should be lesser than the original one (for example, if 950 of 1000 TEs fulfil this condition, the original TE is significant with a 95% of confidence level).

## 4 Results and Discussion

The analysis of the logarithm of Eq. [4] (log posterior) in function on the number of bins provides useful information: a) both time series are long enough to apply the TE and b) the selection of the optimal number of bins $S$ according to the maximum in the log posterior function (see Fig. 3a – b). The log posterior of SAA anomalies (Fig. 3a) increases sharply according to the number of bins considered, reaching a peak (corresponding to the optimal number of bins $S$=5) and then decreasing. Respect to the GSL anomalies series (Fig. 3b), the log posterior also decreases gradually but the maximum is not

so clear. These behaviors indicate a sufficient amount of data to develop this analysis with the TE, but finite sample effects could be important. Due to the lack of an obvious peak in the GSL anomalies series, we establish an agreement between the log posterior curve and the main characteristics of the histogram of the time series. In view of Fig. 3d, we consider that with $S$=4 we have captured the main information of this series. Finally, in order to avoid a future bias in the computation of the TE, we choose the same number of bins $S$ for both time series i.e., equal to 4 (see Table 1) due to larger bin sizes (smaller $S$) are

usually favored in the literature because show the differences more sharply [Sandoval Jr., 2014].

As indicated in the methodology, the selection of the embedding dimension $k$ for both series was carried out using the mutual information given by Eq. [5]. Results are plotted in Fig. S3a – b and contained in the Table S1b. For the GSL anomaly series the optimal dimension was obtained for $k_{GSL}$ = 13, while different values were obtained for the 3 SAA anomalies series (24 for the SAA anomalies series of Jackson et al., 2000; and 26 for the other two series). Nevertheless, since differing

embedding dimensions can generate TE bias [Kraskov et al., 2004] we have fixed the dimension $k_{SAA}$ in 26 for all the 3 SAA anomalies series because a slight over-embedding does not compromise the detection of significant TE [Lindner et al., 2011]. To corroborate the different value of dimension $k$ for GSL and SAA series, we have also calculated the autocorrelation function since the simplest estimate of an optimal $k$ is the first zero of the autocorrelation function [Abarbanel, 1996; Kantz and Schreiber, 1997]. The problem is that these estimates generally yield too large $k$ values for stochastic dynamical systems

[Ragwitz and Kantz, 2002]. In fact, the first minimum reported for the SAA anomalies is given in $k_{SAA}$ = 29 and for the GSL anomalies in $k_{GSL}$ = 17 (Fig. S3c-d). As provided by the mutual information ($k_{SAA}$ = 26 and $k_{GSL}$ = 13), the autocorrelation functions also indicate a lower memory for the GSL series than the three SAA series.





In order to evaluate how the selection of these parameters ($S$, $k$) affects the results, we have performed several tests using different sets of them. The results are detailed in the Supplementary Material along with the Table S1 and S2. We find that the selection of the number of bins and the embedding dimension does not significantly affect our results.

For the chosen parameters, the TE results (Eq. [3]) are given in Table 2 and Figs. 4 and 5. As it can be observed, there is a significant information flow from SAA to GSL anomalies. This finding seems to be independent of the model used to compute the SAA surface extent, which reinforce this result. The significant levels calculated following the IAATF approach are also clarifying, with percentages around the 90% in all cases for the TE from SAA to GSL anomalies. Moreover, this outcome suggests interactions between the two time series of anomalies at a time scale lower or equal to two consecutive data, i.e. one year. This result indicates that the SAA anomalies add predictability to the GSL anomalies and therefore, it would be expected that a future SAA anomaly taking into account our selected trend generates a GSL anomaly with a time lag of one year or less.

Several physical mechanisms are proposed to explain this possible coupling [see De Santis et al., 2012]. The first of them is that an increase of the SAA area facilitates the entrance of charged particles from space. If the SAA area extent grows more than it is expected (positive anomaly), then this entrance is favored.  As a result we have a warmer atmosphere, which implies a consequent melting of major ice caps (Antarctica and Greenland) that finally would cause a greater increasing of the global sea level (positive anomaly). Another mechanism proposed is that a possible reduction of the ozone layer in the upper stratosphere over the South Atlantic region can modify the radiative flux at the top of the atmosphere and hence can cause changes in the weather and climate patterns, including cloud coverage. Finally, an internal mechanism was presented by which a convective dynamism in the outer core could cause a variation of the magnetic field and an elastic deformation at the Earth's surface [Greff-Lefftz et al., 2004].

## 5 Conclusions

We have applied for the first time a recent statistical tool, transfer entropy, to shed light on the question of a possible link between the Earth's magnetic field and climate. In this work we have analyzed two real time series with an analogous evolution for the last 300 years, the South Atlantic Anomaly area extent on the Earth's surface and the Global Sea Level rise. We have analyzed the anomalies of both time series, after removing the long term trend. The results seem to support the existence of an information flow between SAA and GSL anomalies, with larger information transferred from SAA to GSL and a confidence level about 90%. This result has been obtained independently of the model used to compute the SAA surface extent and the selected optimal parameters.

Although this work seems to provide a favorable argument to this link, future investigations are needed to completely exploit this issue, for example to check other time series at longer timescales.



# 6 Acknowledgments

S.A.C. and M.L.O. are grateful to the Spanish research project CGL2014-54112-R of the Spanish Ministerio de Economía y Competitividad and the FPI grant BES-2012-052991, which has allowed S.A.C. two 3-month stays at INGV in Rome in 2014 and 2015 (EEBB-I-14-09023 and EEBB-I-15-10151). A.D.S. and F.J.P.C. also thank the ESA-funded Projects TEMPO and SAFE, for providing partial financial support to this research, and the INGV for providing computational and service facilities. All algorithms have been developed in Matlab codec (Matlab 7.11.0, R2010b) and R-project (R 2.12.2) along with the figures. The data used are listed in the references, tables and main manuscript.

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



# 7 Tables and Figures

**Tables**

**Table 1.** Selection of the optimal number of bins $S$ and the embedding dimension $k$ for SAA and GSL anomalies.

| | OPTIMIZATION PARAMETERS | | | |
|---|---|---|---|---|
| | SAA surface extent | | | GSL |
| | *Jackson et al.* (2000) | *Gubbins et al.* (2006) | *Finlay* (2008) | |
| **S** | 4 | 4 | 4 | 4 |
| **k** | 26 | 26 | 26 | 13 |



**Table 2.** Transfer entropy and statistical significance (in brackets) from SAA to GSL anomalies and from GSL to SAA anomalies, with the optimal parameters (*S* and *k*) reported in the Table 1, and *l*=1.

|  | *Jackson et al.* (2000) | *Gubbins et al.* (2006) | *Finlay* (2008) |
|---|---|---|---|
| $TE_{SAA \rightarrow GSL}$ [bits] | 0.091 (85%) | 0.10 (98%) | 0.11 (99%) |
| $TE_{GSL \rightarrow SAA}$ [bits] | 0.040 (72%) | 0.027 (48%) | 0.027 (48%) |



**Figures**

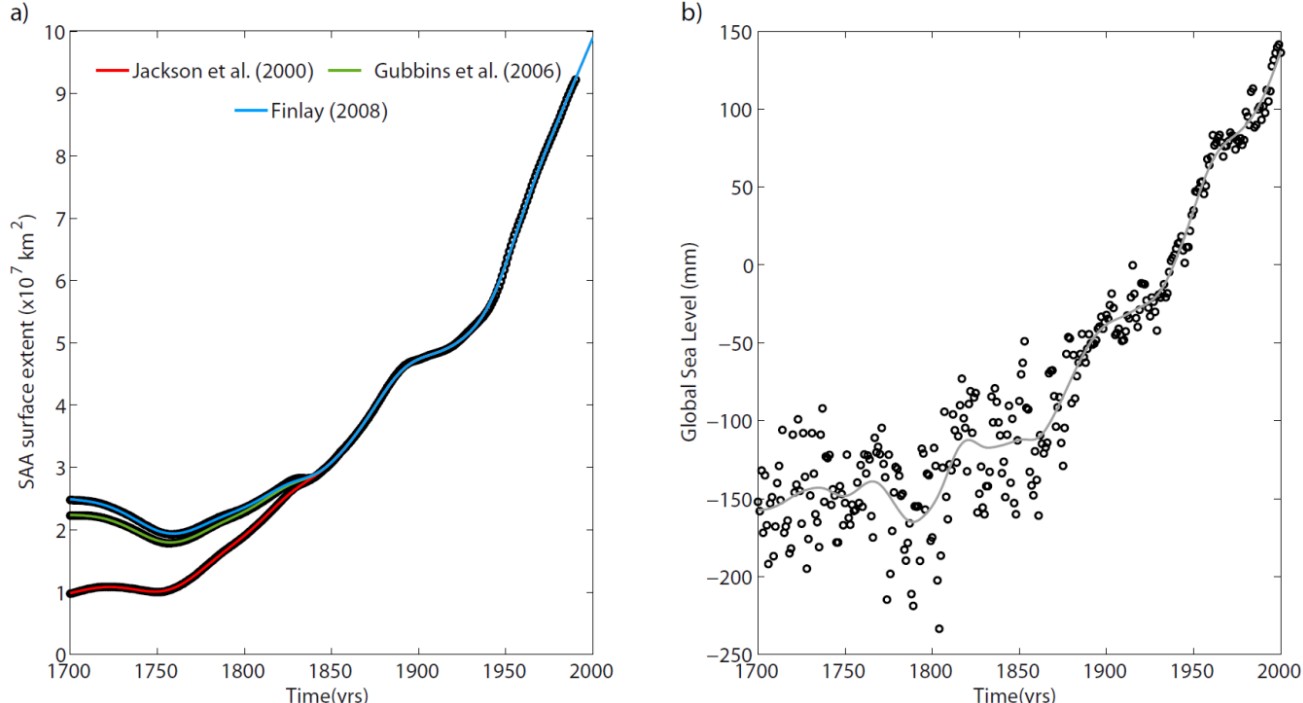

**Figure 1:** Evolution of a) the SAA area extent on the Earth's surface in $10^7$ km$^2$ from three global geomagnetic field models (GUFM1, Jackson et al. [2000] and Gubbins et al. [2006] and Finlay [2008] historical models) and b) GSL rise in mm, for the last 300 years (1700-2000). The lines represent the fits by using penalized cubic splines: (red, green, blue) SAA derived from Jackson et al. [2000], Gubbins et al. [2006] and Finlay [2008], respectively, and (gray) GSL.




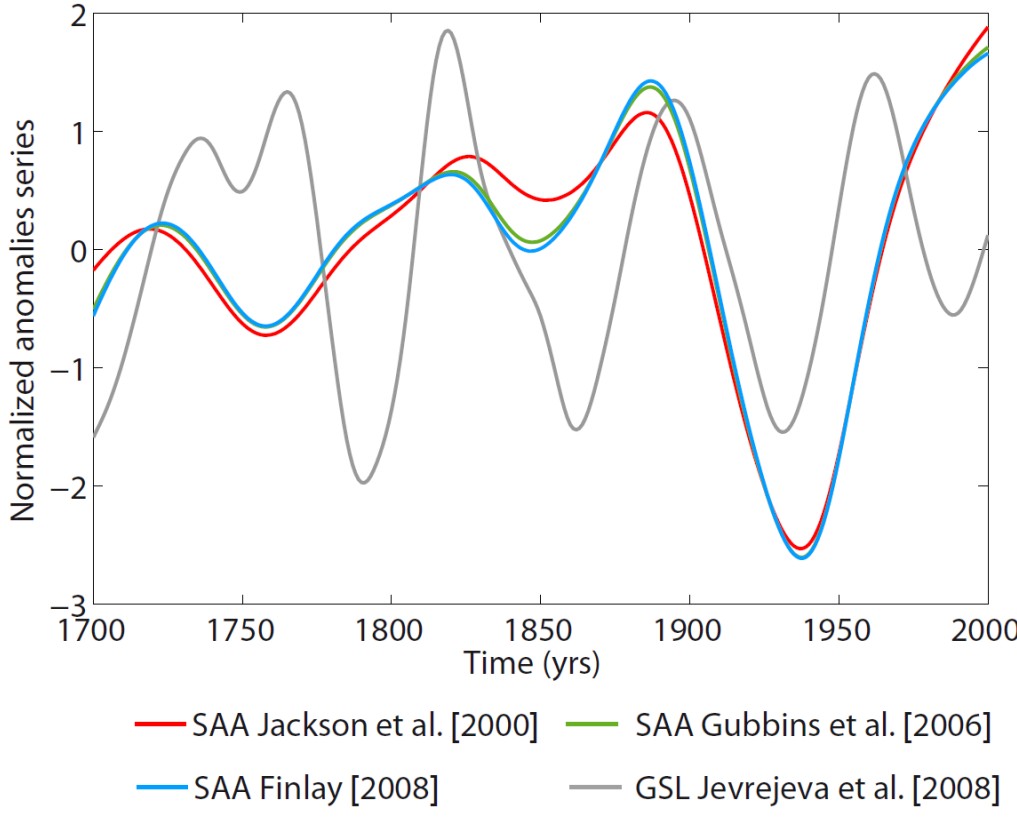

**Figure 2:** Red, green, blue lines correspond to SAA anomalies derived from Jackson et al. [2000], Gubbins et al. [2006] and Finlay [2008], respectively. Grey line represents the GSL anomalies. See text for further details. Both time series have been normalized to zero mean and unit variance.



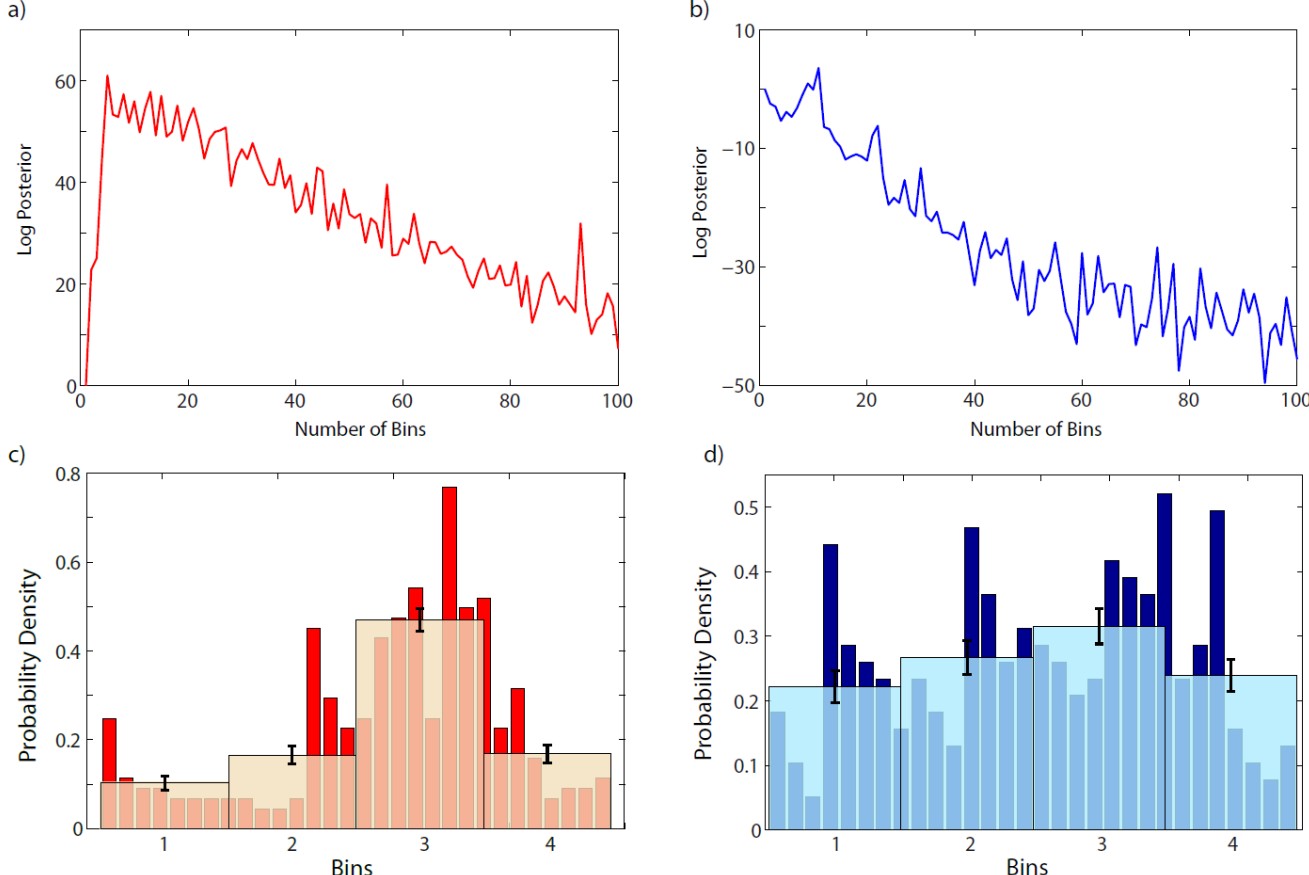

**Figure 3:** Log posterior curves in function on the number of bins S: a) for the SAA anomalies computed from Jackson et al. [2000] and b) for the GSL anomalies. The subplots c) and d) represent, in orange and cyan respectively, the chosen discretization (S=4) taking into account the results given in a) and b), as well as the main characteristics of the probability density of both systems (see red and blue bars in c) and d) plots). The error bars indicate the standard deviation of the bin heights.

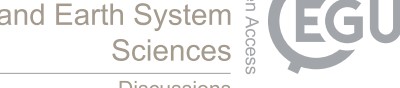



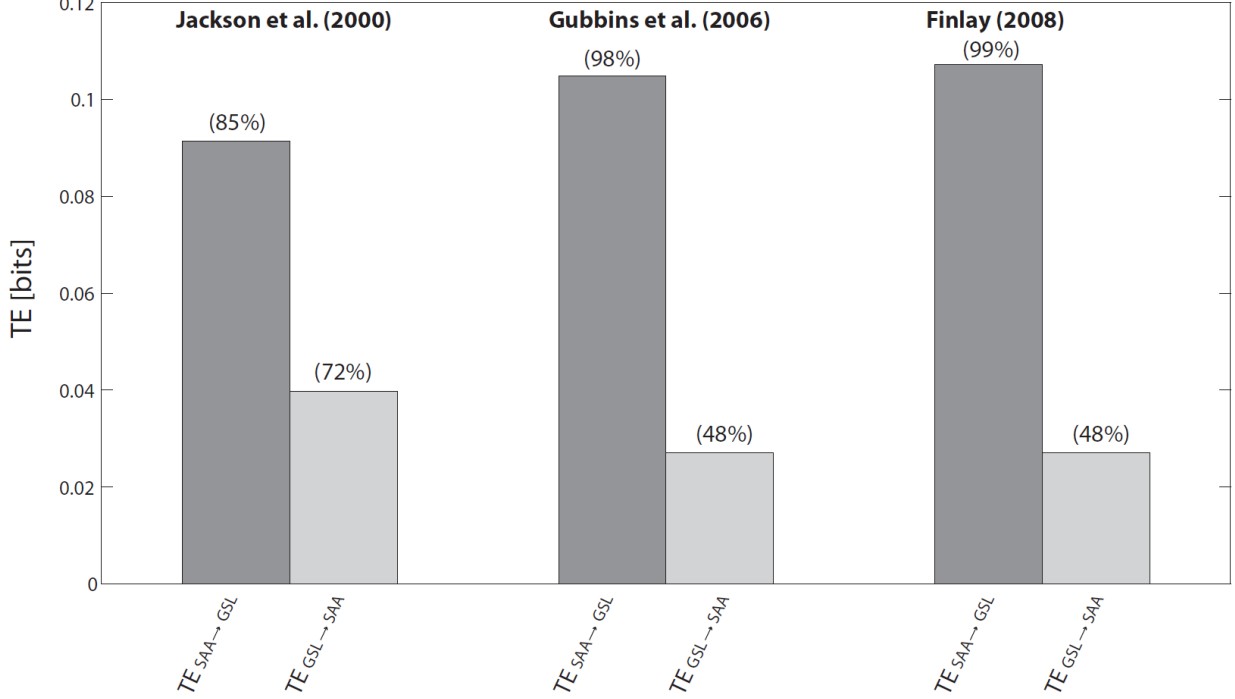

**Figure 4:** Transfer entropy by measuring the information flow from SAA to GSL anomalies and from GSL to SAA anomalies, by using the three historical models for the geomagnetic field to compute the SAA surface extent. In brackets the significant level is shown.





**Figure 5:** Transfer entropy calculated from surrogate series a), c) and e) of SAA anomalies from Jackson et al. [2000] (SAAJ), Gubbins et al. [2006] (SAAG) and Finlay [2008] (SAAF) respectively and b), d) and f) GSL anomalies. The results show that the statistical significance is higher when the sense of the information goes from SAA to GSL anomalies, also registering greater values of the TE.