# Peer review of "Transfer Entropy between South Atlantic Anomaly and Global Sea Level for the last 300 years"

_Natural Hazards and Earth System Sciences, 2016_

## Referee Comment (RC1) · Anonymous Referee #1 · 23 Aug 2016

Report on the paper "Transfer Entropy between South Atlantic Anomaly and Global Sea Level for the last 300 years" by S. A. Campuzano, A. De Santis, F. J. Pavón-Carrasco, M. L. Osete, and E. Qamili. The paper deals with the interesting and important subject regarding a possible effect on climate of a changing magnetic field of the Earth. The paper is a follow-up on the observations of a correlation between the South Atlantic Anomaly (SAA) and the Global Sea Level (GLS) reported in the published paper: "Geomagnetic South Atlantic Anomaly and global sea level rise: A direct connection?" by A. De Santis, E. Qamili, G. Spada, and P. Gasperini. The present paper focuses on the possible causal information link between the anomalies of SAA surface extent and GSL rise for the last 300 years by means of a statistical tool for non-linear dynamic

studies which measures the information flux and the sense of this flux between two systems described by Schreiber (2000). The study concludes that the relationship between the trends in the two time series also exists for shorter timescales and that this within a confidence level of 90 % indicates a cause and effect relationship between the SAA and GSL. Although the methodology and the data selection confirms the results, statistically, I am less convinced about the advancements in the science and understanding of the physical processes that create the two time series postulated to be physically connected. In geophysics many discoveries start by observations and in particular observations of physical parameters that seem to be physically related, for example showing a correlation. I think it is important that such correlations are being communicated to the scientific community also without necessarily indicating a physical mechanism, so that other scientists can contribute with their ideas regarding a physical explanation. However, if such a correlation has been identified, like in the first paper (De Santis et al., 2012), the next step should be to select a physical mechanism, possibly in several steps, where appropriate quantitative relationships between physical parameters can be tested (falsified) using various statistical models. I find it less useful just to apply another statistical tool to verify the already found correlation unless the new statistical tool contradicts (and thereby falsifies) the found result. The present paper promises to apply the results to various proposed physical mechanism but in fact only refers to those already mentioned De by Santis et al., (2012). The first of them is that an increase of the SAA area facilitates the entrance of charged particles from space. If the SAA area extent grows more than it is expected (positive anomaly), then this entrance is favored. As a result we have a warmer atmosphere, which implies a consequent melting of major ice caps (Antarctica and Greenland) that finally would cause a greater increasing of the global sea level (positive anomaly). Another mechanism proposed is that a possible reduction of the ozone layer in the upper stratosphere over the South Atlantic region can modify the radiative flux at the top of the atmosphere and hence can cause changes in the weather and climate patterns, including cloud coverage. However both these proposed mechanisms need to be quantified in

a manner making them available for direct physical test including, for example predictions that can be tested. In the present paper no way forward is presented by which the claimed superiority of the presented statistical tool can be used to distinguish between the proposed mechanism. Therefore I do not see that the present paper represents an advancement in our understanding of the physical mechanisms involved in the claimed relationship between the SAA an the GSL. Without a clear demonstration of how the presented statistical tool can be used to distinguish between the proposed physical mechanisms I am not able to recommend publishing of the paper.

---

## Author Comment (AC1) · 6 Sep 2016

Dear reviewer,

Thank you very much for your comments.

We agree that in this work we analyse the same two time series proposed by De Santis et al. (2012) to study the possible connection between the climate and the geomagnetic field. However, there are two new aspects which are worth pointing, that indicate this paper is an important advance with respect the previous one.

The first aspect is that, while in the 2012 paper the authors studied the possible correlation on the long trend of the time series, in the present work we filter this long trend

and analyse shorter scales. From this point of view, our study completes the previous one, confirming the connection.

The second aspect, and this is the most important part of the paper (and justifies the given title), is that we have used a new statistical tool that is able to measure the independence between two time series, and, in case of some dependence, also the direction of the information flow. This concept is important and new in the field. Transfer entropy does not simply establish correlations. For example, if two time series were completely correlated, the reported value of the transfer entropy could be zero. This means that both series are independent, i.e. the knowledge of one of this series does not improve the knowledge of the other one. Otherwise, when a series is depending on the other, the transfer entropy provides a measure of this dependence, together with the flow of information, i.e. knowing the behaviour of one time series makes possible a reliable prediction on the evolution of the other one. Therefore, the use of the transfer entropy provides a new dimension in the study of the connection between the climate and the geomagnetic field, because it implies, if there exists, an information flow between the two time series, and is able to distinguish the sense of this flow, an innovative property of the method.

The reviewer points out that "the proposed mechanisms need to be quantified in a manner making them available for direct physical test including, for example predictions that can be tested. In the present paper no way forward is presented by which the claimed superiority of the presented statistical tool can be used to distinguish between the proposed mechanism". The reviewer also claims that we do not provide "a clear demonstration of how the presented statistical tool can be used to distinguish between the proposed physical mechanisms in the paper". We do not agree with him/her. The response is in the capacity of the transfer entropy to distinguish the sense of the information flow. In the analysed case study, we have shown that the sense of the information goes from SAA to GSL time series. This would discard any physical mechanism in which the climate controls the geomagnetic field and support the mechanisms caused

by the presence of the SAA. In addition, the outcome of this work suggests interactions between the two time series of anomalies, not only at the long-term trend, but also at a time scale lower or equal to one year. This indicates that, if we are right, the SAA anomalies add great predictability to the GSL anomalies and therefore, it would be expected that a future SAA anomaly (taking into account our selected trend) generates a GSL anomaly with a time lag of one year or less. Hence, any physical mechanism proposed to explain this relation should act within this time interval, excluding many other mechanisms with longer time lags. In order to clarify these points in the manuscript, we propose adding this paragraph in the Discussion section of the paper.

To summarise, we cannot yet establish which the physical mechanism that explains this connection is, but we believe that we are able to point out, with a 90% of confidence level, the sense of this mechanism (and the time interval in which should act), and this is an important advance in the field. Our study is a significant step forward in understanding the complex phenomenon that produces the present increase of GSL, and its possible connection with the present geomagnetic field, characterised by a comparable complexity.

---

## Referee Comment (RC2) · Anonymous Referee #2 · 11 Apr 2017

The paper explores potential relationships between the spatial extension of South Atlantic Anomaly SAA, a feature of the changing Earth's magnetic field, and the Global Sea Level GSL. The time period considered consists of the recent 300 years is. In this period, the magnetic dipole field of the Earth has been declining. The increasing (and westward moving) magnetic anomaly at the ocean surface over the South Atlantic is an important ingredient in this process (see, for example, Finlay et al., Nature, 2016). At the same time, Global Sea Level has been increasing. The parallel development of the phenomena has been discussed in a paper of De Santis et al., JASTP, 2012. Two of the current authors (De Santis and Qamili) have also been authors of the former paper.

While the 2012 paper focused on the long term trends, the current paper and seeks to

investigate the shorter term variations beyond the nonlinear trend. For this purpose, the time series were smoothed using penalized cubic splines, and the long term trends we removed using 2nd order polynomials. A procedure called transfer entropy is applied in order to estimate the statistical relationship. It is stated that there is a significant relationship with a lag of one year or less, the South Atlantic Anomaly leading the Global Sea Level variations.

Considering two parallel trends without a physical explanation is generally a problematic approach. The supposed relationship of cosmic rays (with their intensity influenced by the magnetic field) and clouds (which are again affecting temperatures) has amply been discussed in the recent IPCC WG1 report. The synthesis given there is as follows: "Although there is some evidence that ionization from cosmic rays may enhance aerosol nucleation in the free troposphere, there is medium evidence and high agreement that the cosmic ray-ionization mechanism is too weak to influence global concentrations of CCN or droplets or their change over the last century or during a solar cycle in any climatically significant way. " The manuscript doesn't come up with another process that could physically explain the relationship (I do not consider a suspected regional O3 change a credible link between SAA and GSL). Still, a thorough demonstration of the characteristics of the relationships on shorter time scales may be of some value in order to point at close statistical links beyond a common trend in both time series.

Regretfully, the authors fail to produce a convincing strategy in this respect. Other than Finlay et al., 2016, they use data before 1840 (begin of the era of direct geomagnetic observations). These are apparently not covered by observations, and thus may not be reliable. Still, this part of the data represents almost half of the complete time series and thus has an obvious influence on the subsequent statistical analysis. The authors use a complicated methodology involving the estimation of additional parameters for demonstrating the existence of a relationship. In doing so, they fail to provide the information that could be used to point at underlying physical processes, which must be the

intention. Looking at Fig. 2, I have doubts about the existence of a stable relationship between the parameters considered. I would also ask about the relationship in different frequencies which may hint at a relationship.

Solving these issues would considerably change the paper, and thus I recommend to reject it. In addition, I think that the paper is not particularly suitable for publication in NHESS as it is not considering a specific natural hazard.